# 3D-AffordanceLLM: Harnessing Large Language Models for Open-Vocabulary Affordance Detection in 3D Worlds

**Hengshuo Chu[1], Xiang Deng[†1], Qi Lv[1], Xiaoyang Chen[1], Yinchuan Li[2], Jianye Hao[2], Liqiang Nie[†1]**
[1]Harbin Institute of Technology (Shenzhen), [2]Huawei Noah's Ark Lab

## Abstract

3D Affordance detection is a challenging problem with broad applications on various robotic tasks. Existing methods typically formulate the detection paradigm as a label-based semantic segmentation task. This paradigm relies on predefined labels and lacks the ability to comprehend complex natural language, resulting in limited generalization in open-world scene. To address these limitations, we reformulate the traditional affordance detection paradigm into *Instruction Reasoning Affordance Segmentation* (IRAS) task. This task is designed to output a affordance mask region given a query reasoning text, which avoids fixed categories of input labels. We accordingly propose the *3D-AffordanceLLM* (3D-ADLLM), a framework designed for reasoning affordance detection in 3D open-scene. Specifically, 3D-ADLLM introduces large language models (LLMs) to 3D affordance perception with a custom-designed decoder for generating affordance masks, thus achieving open-world reasoning affordance detection. In addition, given the scarcity of 3D affordance datasets for training large models, we seek to extract knowledge from general segmentation data and transfer it to affordance detection. Thus, we propose a multi-stage training strategy that begins with a novel pre-training task, i.e., *Referring Object Part Segmentation* (ROPS). This stage is designed to equip the model with general recognition and segmentation capabilities at the object-part level. Then followed by fine-tuning with the IRAS task, 3D-ADLLM obtains the reasoning ability for affordance detection. In summary, 3D-ADLLM leverages the rich world knowledge and human-object interaction reasoning ability of LLMs, achieving approximately an 8% improvement in mIoU on open-vocabulary affordance detection tasks.

## 1 Introduction

Robots are increasingly integrating into various aspects of our daily life (Matheson et al., 2019). As we progress toward developing the next generation of more advanced robotic agents, it is essential to enable robots to comprehend natural language instructions within context and to perceive task-relevant information in their surroundings. This skill is particularly vital for seamless interactions in unstructured environments, such as homes, where adaptability to diverse situations is crucial. Specifically, the robots need to not only identify the objects in the environments but also locate the specific regions of each object that are suitable for interaction: *affordance*.

The concept of affordance was introduced by ecological psychologist James Gibson (Gibson, 1966) and has since played a significant role in various robotic applications, including object recognition (Hong et al., 2023a; Hou et al., 2021), action anticipation (Roy & Fernando, 2021), agent activity recognition (Chen et al., 2023), and object functionality understanding (Li et al., 2023). Affordance describes potential interactions between robots and their environment, such as using a knife's blade for cutting tasks. Detecting affordances is challenging due to object diversity and complexity (Min et al., 2016). Traditionally, 2D images and CNNs are used (Nguyen et al., 2016; Do et al., 2018; Pacheco-Ortega & Mayol-Cuervas, 2022)(Krizhevsky et al., 2012), but 2D information

---

[†]Corresponding authors

lacks the depth necessary for precise manipulation, necessitating 3D transformations (Deng et al., 2021).

With advanced depth cameras, 3D point clouds have become a widely used modality in robotic applications (Liu et al., 2019). Unlike conventional images, 3D point clouds offer robots direct and detailed 3D information about surrounding objects and environments. Hence, the 3D affordance detection has been deemed as a critical step in bridging perception and manipulation in the physical world for an embodied agent, thus has shown substantial impact on practical applications such as robotic manipulation (Geng et al., 2023; Moldovan et al., 2012). While current approaches are limited by fixed label sets (Deng et al., 2021; Mo et al., 2022), reducing flexibility and generalization in dynamic settings. To overcome the fixed label set problem in affordance detection, Nguyen et al. (Nguyen et al., 2023) have incorporated a text encoder to enable models to handle certain levels of open-vocabulary detection, but these algorithms still rely on a classification based training paradigm. As a result, they lack the ability for rapid and continuous learning when presented with new affordance label data. Furthermore, current affordance detection methods also heavily rely on the predefined labels and lack the ability to understand and reason over long contextual text. Additionally, the scarcity of 3D affordance datasets (Deng et al., 2021; Nguyen et al., 2023) constrains the effective training of large-scale models.

Towards these issues, we redefine the 3D affordance detection as an *Instruction Reasoning Affordance Segmentation* (IRAS) task and accordingly propose *3D-AffordanceLLM* (3D-ADLLM). The IRAS task is designed to output an affordance mask region in response to complex, reasoning-based query text, overcoming the limitations of fixed affordance labels and the difficulty of understanding complex instructions. Our 3D-ADLLM framework introduces large language models (LLMs) to 3D affordance perception with a specifically designed decoder for generating affordance masks, thus achieving open-world reasoning affordance detection. Specifically, we introduce an additional token, <AFF>, into the original LLM vocabulary. When the <AFF> token is generated, its hidden embedding is further decoded into the corresponding segmentation mask. By representing the segmentation mask as an embedding, 3D-ADLLM not only gains segmentation capability but also benefits from end-to-end training. However, due to the scarcity of 3D affordance datasets for training large models, we propose a multi-stage training strategy to extract knowledge from general segmentation data and transfer it to affordance detection. This process involves pre-training on PartNet (Mo et al., 2019) with *Referring Object Part Segmentation* (ROPS) tasks to acquire the object-part level general recognition and segmentation knowledge. Subsequently, we fine-tune the model with the IRAS task to achieve context-aware reasoning ability and robust performance in open-set zero-shot affordance detection.

Our main contributions are summarized as follows:

- Different from the existing affordance detection methods that rely on fixed sets of labels, we address this limitation by introducing a new detection paradigm based on the *Instruction Reasoning Affordance Segmentation* (IRAS) task. By reforming the label-based semantic segmentation task in the traditional affordance detection paradigm into a natural language-driven reasoning affordance segmentation task, our model enables more flexible and context-aware reasoning, facilitating effective zero-shot learning capabilities.

- To address the IRAS tasks driven by semantic complex natural language, we consequently propose the *3D AffordanceLLM* (3D-ADLLM) model, combining a large language model (LLM) with a carefully designed Affordance Decoder. Our 3D-ADLLM framework can understand semantically-rich, long-context instructions and leverages the LLM's world knowledge for superior open-vocabulary affordance detection.

- Due to the scarcity of 3D affordance datasets for training large models, we propose a multi-stage training strategy to transfer general segmentation knowledge into affordance detection. First, the model is equipped with general recognition and segmentation knowledge through a novel pretraining task, i.e., the Referring Object Part Segmentation (ROPS). Subsequently, the model is fine-tuned with the IRAS task to handle context-aware reasoning and affordance region prediction.

## 2 RELATED WORK

**Affordance Detection.** Originating from the 2D domain, initial work in affordance detection primarily focused on identifying objects with affordances (Do et al., 2018). Building on this foundation, later studies (Lu et al., 2022) introduced linguistic descriptions to improve detection, but they continued to emphasize object-level affordances, lacking fine-grained analysis. Addressing this problem, subsequent research (Chen et al., 2023; Li et al., 2023; Luo et al., 2022; Nagarajan et al., 2019; Mi et al., 2020) has focused on detecting specific affordance parts, establishing a new benchmark for precision in the field. With the advancement of embodied AI, the scope of affordance learning has expanded into 3D domain. 3D AffordanceNet (Deng et al., 2021) introduces the first benchmark dataset for learning affordance from object point clouds. IAGNet (Yang et al., 2023) propose a setting for learning 3D affordance parts guided by image queries. Recently, some work (Nguyen et al., 2023) also explores the open-vocabulay affordance detection in point clouds. However, these methods primarily focus on linking object geometric features with fixed affordance labels, overlooking the semantic aspect. This limitation makes it challenging to understand natural language instructions and hampers the ability to generalize affordance detection to unseen scenarios. In contrast, the proposed 3D-ADLLM overcomes the limitations of fixed label sets and enhance the ability to comprehend semantic complex description. Specifically, we shift the detection paradigm from label-based semantic segmentation into Instruction Reasoning Affordance Segmentation (IRAS).

**3D Large Multi-Modal Models.** 3D object-level LMMs (Yu et al., 2022; Xue et al., 2023; Zhou et al., 2023) have successfully bridged the gap between 3D vision and text by leveraging large-scale 3D object datasets like (Deitke et al., 2023; Vishwanath et al., 2009). ShapeLLM (Qi et al., 2024) further advances the embodied interaction and referring expression grounding through its novel and powerful point encoder. However, despite these advances, such models still face challenges in interpreting complex spatial relationships within 3D scenes. For scene-level LMMs, models like Chat-3D (Wang et al., 2023) and LL3DA (Chen et al., 2024) enable interaction with scene objects using pre-selection mechanisms. Building on this foundation, Chat-3D v2 (Huang et al., 2023) enhances referencing and grounding accuracy by incorporating object identifiers, while 3D-LLM (Hong et al., 2023b) improves scene comprehension by integrating positional embeddings and location tokens. Unlike previous works that primarily focus on 3D grounding and understanding, our method introduces a specialized token, `<AFF>`, which enables LLMs to directly detect affordances and generate affordance masks within 3D open-world scene.

## 3 METHOD

### 3.1 PARADIGM REFORMULATION

Affordance detection aims to identify specific regions of objects that are suitable for interaction. It has been deemed as a critical step in bridging perception and manipulation in the physical world for embodied agents. As illustrated in Fig. 1 (a), the traditional paradigm uses a shared point backbone (Qi et al., 2017; Zhao et al., 2021; Wang et al., 2019) to extract point-wise features, and generates masks with a predefined type semantic segmentation head. Alternatively, they leverage a text encoder like CLIP (Radford et al., 2021) to associate point-wise features with text embeddings of affordance labels using cosine similarity, achieving limited open-vocabulary detection on the phrase level. This paradigm relies on predefined labels and has a limited ability to understand complex natural language, which restricts its generalization in 3D open-world scene.

To address these limitations, we introduce a new paradigm formulated as an **I**nstruction **R**easoning **A**ffordance **S**egmentation (**IRAS**) task as depicted in Fig. 1 (b). This paradigm is designed to establish a robust connection between language context and object affordance, avoiding the overreliance on auxiliary affordance label prediction. This approach facilitates a significant improvement in our ability to understand and interact with the physical world.

**IRAS Definition.** *Given a query reasoning instruction $Q_a$ and an object point cloud $P_c \in \mathbb{R}^{n \times 3}$ with N points, the goal of IRAS is to predict a binary mask of $M_a \in \mathbb{R}^N$ that delineates the functional regions pertinent to the query, affordance regions:*

$$F_{Model}(Q_a, P_c) \Rightarrow M_a$$

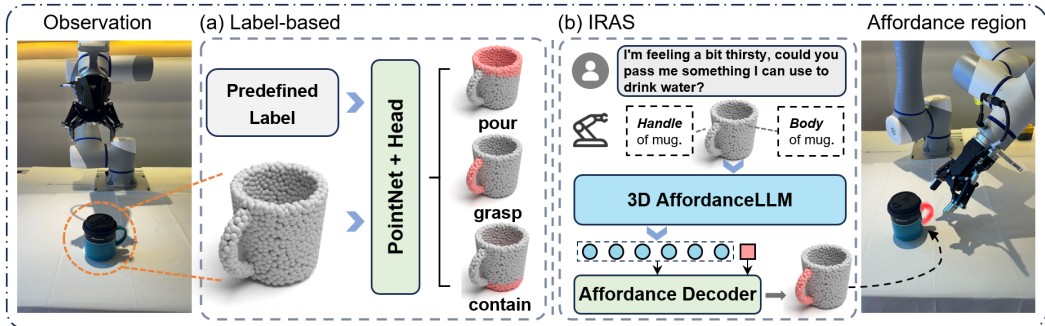

Figure 1: The comparison of the affordance detection paradigm based on our IRAS or traditional label-based segmentation tasks. (a) shows that label-based paradigm can only detect the fixed set of affordance regions through the predefined label and seg-head; (b) demonstrates the IRAS based paradigm forges a link between semantic complex instruction and object affordance, enabling open-world reasoning affordance detection.

## 3.2 3D-AFFORDANCELLM

To the traditional methods that rely on fixed label sets and are limited to short-text detection, IRAS demands robust language comprehension and reasoning to associate the potential affordance in input query with 3D objects areas. Thus, we incorporate large language models (LLMs) into 3D affordance perception. LLMs, trained on trillions of tokens, excel in understanding and reasoning about instructions and possess extensive world knowledge. For instance, when asked where to interact with a mug to grasp it, LLMs suggests using the handle for a firm grip to avoid spilling. This demonstrates LLMs' world knowledge and the capability in understanding human-object interactions. To harness this capability for 3D affordance perception, we introduce the 3D AffordanceLLM Model, aiming to improve affordance detection in previously unseen contexts.

Our framework, *3D AffordanceLLM*, as illustrated in Fig. 2, primarily consists of two main components: (1) a point cloud multimodal model which is trained to accept point cloud and text inputs and generate response, including a special token, <AFF>; (2) an Affordance Decoder (AFD), which extracts hidden layer features from these <AFF> tokens and combines them with segmentation point features to generate affordance masks.

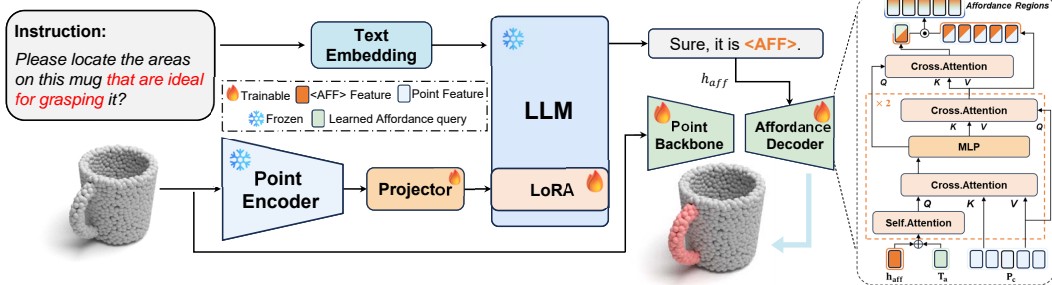

Figure 2: The Pipeline of 3D-ADLLM. Given the input point cloud and query reasoning instruction, the point cloud multimodal model is trained with lora to predict special token <AFF>. Finally, the special token and dense point features from $f_{\mathrm{PB}}$ is fed into our designed affordance decoder to generate the final affordance mask.

### 3.2.1 MODEL ARCHITECTURE

As is shown in Fig. 2, our 3D AffordanceLLM consists of the following modules: a pre-trained point cloud encoder $f_{\mathrm{pe}}$, a projector $f_{\mathrm{proj}}$, a point backbone $f_{\mathrm{PB}}$, an affordance decoder $f_{\mathrm{AFD}}$ and a pre-trained large language model (LLM) backbone $f_{\mathrm{llm}}$.

**Point Encoder.** The point cloud encoder $f_{\text{pe}}$, takes a point cloud $\mathbf{P}_{\text{cloud}} \in \mathbb{R}^{n \times d}$ as input, where $n$ represents the number of points and $d$ denotes the feature dimension of each point. The output of the encoder is a sequence of point features $X = (x_1, x_2, \ldots, x_{\text{m}}) \in \mathbb{R}^{m \times c}$, where $m$ is the number of point features and $c$ is the feature dimension. Similarly, the point backbone $f_{\text{PB}}$, also processes input point cloud $\mathbf{P}_{\text{cloud}} \in \mathbb{R}^{n \times d}$, extracting the dense point cloud features $X' = (x'_1, x'_2, \ldots, x'_{\text{n}}) \in \mathbb{R}^{n \times c'}$, specifically tailored for segmentation tasks. These features are subsequently fed into the Affordance Decoder.

**LLM Projector.** The projector $f_{\text{proj}}$ is a MLP layer that maps the point features $X$ to point tokens $Y = (y_1, y_2, \ldots, y_m) \in \mathbb{R}^{m \times c''}$, where $c''$ is the dimension of the point tokens, matching the dimension of the text tokens.

**Large Language Model.** The LLM backbone $f_{\text{llm}}$ is a decoder-only Transformer model (Vaswani et al., 2017), which processes a sequence of tokens comprising text and point tokens. This mixed token sequence is denoted as $Z = (z_1, z_2, \ldots, z_k) \in \mathbb{R}^{k \times c''}$, where $k$ is the total number of tokens. Leveraging a self-attention mechanism, the LLM backbone captures contextual relationships between different token types, enabling it to generate responses based on both text and point cloud inputs. Formally, the output of the LLM backbone $f_{\text{llm}}$ is a sequence of predicted tokens $\hat{Z} = (\hat{z}_1, \hat{z}_2, \ldots, \hat{z}_k) \in \mathbb{R}^{k \times c''}$. The prediction of the $i$-th token, $\hat{z}_i$, is conditioned on all previous tokens, $Z_{<i} = (z_1, \ldots, z_{i-1})$, which can be expressed mathematically as:

$$\hat{z}_i = f_{\text{llm}}(Z_{<i}).$$

Each $\hat{z}_i$ is passed through a final linear layer followed by a softmax operation, which maps the hidden states to a probability distribution over the vocabulary. This layer is denoted as $f_{\text{vocab}} : \mathbb{R}^{c'} \to \mathbb{R}^V$, where $V$ is the size of the vocabulary. The final prediction $\tilde{z}_i$ for the $i$-th token is the word in the vocabulary with the highest probability, expressed as:

$$\tilde{z}_i = \arg \max_{w \in \text{vocab}} f_{\text{vocab}}(\hat{z}_i)[w].$$

**Affordance Decoder.** Building on the success of learnable query-based methods in object segmentation, we introduce an Affordance Decoder Module (AFD) that leverages a set of learnable output queries conditioned on input questions, termed *affordance queries* $T_{\text{a}}$ to decode segmentation masks. A two-layer decoder updates both the point features and the question features via cross-attention. Then, the updated query tokens and point features are used to dynamically predict affordance masks.

### 3.2.2 EMBEDDING AS AFFORDANCE

Unlike conventional tasks such as grounding, question answering, etc., within the realm of 3D large multi-modal models (LMMs), the IRAS task is depicted to generate a affordance segmentation mask directly given a reasoning query. Most current 3D LLM (such as 3D-LLM (Hong et al., 2023a), ShapeLLM (Qi et al., 2024) support 3D scenes or objects and text as input, but they can only output text or bbox and cannot directly output fine-grained segmentation masks. Inspired by the LISA model (Lai et al., 2024), which directly outputs the segmentation mask in the 2D domain, we adopt a similar idea in 3D affordance detection. To achieve that, we propose the embedding-as-affordance paradigm to inject new affordance segmentation capabilities into the 3D LMM. The pipeline of our method is illustrated in Fig. 2. Specifically, we expand the original LLM vocabulary by adding a new token, <AFF>, which signals a request for an affordance output. Given a complex reasoning instruction query $\mathbf{Q}_{\text{aff}}$ and a point cloud input $\mathbf{P}_{\text{cloud}}$, we feed them into the multimodal point clouds LLM $F_{\text{3D-ADLLM}}$, which outputs a text response $\hat{\mathbf{y}}_{\text{txt}}$: "Sure, it is <AFF>." This process can be formulated as:

$$\hat{\mathbf{y}}_{\text{txt}} = F_{\text{3D-ADLLM}}(\mathbf{P}_{\text{cloud}}, \mathbf{Q}_{\text{aff}}).$$

When the LLM intends to generate a binary affordance mask, the output $\hat{\mathbf{y}}_{\text{txt}}$ would include a <AFF> token. We then extract the LLM last-layer embedding $\tilde{\mathbf{h}}_{\text{aff}}$ corresponding to the <AFF> token and apply an MLP projection layer $\text{Proj}$ to obtain $\mathbf{h}_{\text{aff}}$. Simultaneously, the point cloud backbone $f_{\text{PB}}$ extracts the dense point clouds features $\mathbf{f}$ from the points input $\mathbf{P}_{\text{cloud}}$. Finally, $\mathbf{h}_{\text{aff}}$ and $\mathbf{f}$ are fed to the decoder $f_{\text{AFD}}$ to produce the final affordance mask $\hat{\mathbf{M}}_{\text{aff}}$. The process can be formulated as

$$\mathbf{h}_{\text{aff}} = \text{Proj}(\tilde{\mathbf{h}}_{\text{aff}})$$

$$\mathbf{f} = f_{\text{PB}}(\mathbf{P}_{\text{cloud}}), \hat{\mathbf{M}}_{\text{aff}} = f_{\text{AFD}}(\mathbf{h}_{\text{aff}}, \mathbf{f}).$$

Figure 3: Multi-stage training strategy. Illustration of transferring general segmentation knowledge to affordance detection. (a) depicts the process of extracting general segmentation knowledge, while (b) illustrates the framework for transferring this knowledge to affordance detection

## 3.3 MULTI-STAGE TRAINING

Existing 3D affordance datasets, such as 3D AffordanceNet datasets, OpenAD datasets in (Deng et al., 2021; Nguyen et al., 2023), are constrained in availability and dataset sizes. Thus, given the scarcity of 3D affordance datasets for training large models, we devise a multi-stage training strategy which extracts knowledge from general segmentation data and transfers it to IRAS affordance detection. In addition, due to the varying scales of target affordance regions, we propose a sample unbalanced loss factor to enhance the model's learning effectiveness and adaptability across different region scales.

### 3.3.1 EXTRACTING GENERAL SEGMENTATION KNOWLEDGE

Considering the limited amounts of affordance datasets for training large models, this stage aims to leverage general datasets to equip the model with general recognition and segmentation capabilities at the object-part level. Thus, we introduce **R**eferring **O**bject **P**art **S**egmentation (**ROPS**) task to acquire the general knowledge.

**ROPS Definition.** *Given a referring expression that denotes the name of the object's components $Q$ and an object point cloud $P_c \in \mathbb{R}^{n \times 3}$ consisting of N points, the objective of ROPS is to predict a binary mask for $M_p \in \mathbb{R}^N$ that corresponds to the query:*

$$F_{Model}(Q_p, P_c) \Rightarrow M_p$$

In the pre-training phase, we employ the framework in Fig. 3 (a) to train the ROPS task on the PartNet dataset (Mo et al., 2019). As depicted in Fig. 3 (a), the object point cloud is processed by a trainable backbone to extract point features $\mathbf{f}_{P_{cloud}}$.The object part descriptions are encoded using a frozen text encoder to generate text features $\mathbf{f}_{Q_{part}}$, which are then mapped via an offline MLP layer to produce $\mathbf{f}'_{Q_{part}}$. Finally, $\mathbf{f}'_{Q_{part}}$ and $\mathbf{f}_{P_{cloud}}$ are passed into the Mask Decoder to generate the final part mask $\mathbf{M}_{part}$, formulated as:

$$\mathbf{M}_{part} = \text{MaskDecoder}(\mathbf{f}'_{Q_{part}}, \mathbf{f}_{P_{cloud}})$$

**Training Objectives.** Unlike (Mo et al., 2019), which uses a multi-class head for prediction, our stragety seeks to extract the knowledge relationship between referring object part description and the corresponding mask regions. Thus, we solely employ Dice Loss and Binary CrossEntropy (BCE) loss to guide the segmentation mask prediction.

$$\mathcal{L} = \lambda_1 \mathcal{L}_{BCE} + \lambda_2 \mathcal{L}_{DICE}.$$

### 3.3.2 TRANSFERRING KNOWLEDGE TO AFFORDANCE DETECTION

Building upon the extensive segmentation knowledge acquired from the ROPS task, we transfer this knowledge to affordance detection by IRAS finetuning to enhance the model's generalization. We

also propose a sample unbalanced loss factor to address the learning strategies for affordance regions of different scales. Specifically, during IRAS fine-tuning: we use the pretrained checkpoint $W_{f_{PB}}$ and $W_{f_{MD}}$ to initialize the modules $f_{PB}$ and $f_{AFD}$ in our framework 3D-ADLLM as shown in Fig. 2. We then use the Lora method to fine-tune a pre-trained LLM for affordance segmentation.

**Training Objectives.** The model is trained end-to-end using text generation loss $\mathcal{L}_{txt}$ and segmentation mask loss $\mathcal{L}_{mask}$. Specifically, $\mathcal{L}_{txt}$ encourage the LLMs generate the response including the `<AFF>` token and forces the features to map to the same `<AFF>` placeholder. Then, $\mathcal{L}_{mask}$ diversifies the `<AFF>` features to contain the information for affordance prediction and guides affordance mask generation. The overall objective $\mathcal{L}$ is the weighted sum of these losses, determined by $\lambda_{txt}$ and $\lambda_{mask}$:

$$\mathcal{L} = \lambda_{txt}\mathcal{L}_{txt} + \lambda_{mask}\mathcal{L}_{mask}.$$

Specifically, $\mathcal{L}_{txt}$ is the auto-regressive cross-entropy loss for text generation, and $\mathcal{L}_{mask}$ is the mask loss for high-quality segmentation. To compute $\mathcal{L}_{mask}$, we use a combination of per-pixel BCE loss and DICE Loss, with weights $\lambda_{bce}$ and $\lambda_{dice}$. Given the ground-truth targets $\mathbf{y}_{txt}$ and M obtained from dataset, these losses are formulated as:

$$\mathcal{L}_{txt} = \mathbf{CE}(\hat{\mathbf{y}}_{txt}, \mathbf{y}_{txt}),$$

$$\mathcal{L}_{mask} = \lambda_{bce}\mathbf{BCE}(\hat{\mathbf{M}}, \mathbf{M}) + \lambda_{dice}\mathbf{DICE}(\hat{\mathbf{M}}, \mathbf{M}).$$

**Sample Unbalanced Loss Factor.** Due to the varying scales of target affordance regions, our model 3D-ADLLM naturally challenges model's adaptiveness at different scales. This variability will results in an imbalance in the difficulty of learning samples between different affordance types during the training process. To mitigate the issue of sample imbalance across different affordance types during training, we apply weights to the mask losses for each class. The weighted loss is defined as: $\mathcal{L}_{mask} = \sum_{i=1}^{n} \omega_i \mathcal{L}_{mask}^i$. The weight $\omega_i$ is calculated as:

$$\omega_i = \left( \frac{\max\{c_1, c_2, \ldots, c_m, c_0\}}{c_i} \right)^{1/4}$$

where $c_i$ is the number of ground truth points for class $i$, and $c_0$ denotes background points.

# 4 EXPERIMENT

## 4.1 EXPERIMENT SETTING

**Network Architecture.** We use Phi-3.5-mini-instruct ($f_{llm}$) (Abdin et al., 2024) as our base LLM. For the point encoder ($f_{pe}$), we adopt Point-BERT (Yu et al., 2022), pre-trained with ULIP-2 (Xue et al., 2024) in the ModelNet dataset (Vishwanath et al., 2009). The projector layer ($f_{proj}$) between the point encoder $f_{pe}$ and the LLM $f_{llm}$ is a linear layer. Additionally, we utilize the Point Transformer (Zhao et al., 2021)as the backbone for our point segmentation model ($f_{PB}$).

**Datasets.** As is mentioned in Sec. 3.3, our training data is made up of two types of task data: (1) *Referring Object Part Segmentation Dataset:* we build this dataset on PartNet (Mo et al., 2019), which contains 573,585 part instances across 25,571 3D models and 24 object categories. For pre-training, we split it into single-part segmentation instances. (2) *Instruction Reasoning Affordance Segmentation Dataset:* we meticulously compile a question-point affordance dataset with 42119 paired samples from 3D AffordanceNet dataset (Deng et al., 2021), covering 23 classes and 36 affordance types. The Detailed data settings (full or partial view) and the visualized data analysis can be seen in Appendix Sect. A.3.

**Baseline Models.** We compare our method with the following recent methods for zero-shot learning in 3D point clouds: ZSLPC (Cheraghian et al., 2019), TZSLPC (Cheraghian et al., 2020), 3DGenZ (Michele et al., 2021), OpenAD (Nguyen et al., 2023), IAGNet (Yang et al., 2023), LASO (Li et al., 2024) and ShapeLLM (Qi et al., 2024). Detailed baseline model explanation for experiments can be found in Appendix Sect. A.1.

**Evaluation metrics.** We divide the IRAS dataset following the split of OpenAD and evaluate the close-set and open-set of IRAS. According to Nguyen et al. (2023), we use three metrics to evaluate the results over all **classes**: $mIoU^c$ (mean IoU over all classes), $Acc^c$ (overall accuracy over all

Table 1: Main results of 3D-ADLLM on zero-short open vocabulary detection. The result is calculated over all classes. The overall results of all comparative methods, the best results are in bold. $*$ The method of ShapeLLM is tested without finetuning.

| Method | Full-view | | | Partial-view | | |
|---|---|---|---|---|---|---|
| | mIoU$^c$ | Acc$^c$ | mAcc$^c$ | mIoU$^c$ | Acc$^c$ | mAcc$^c$ |
| TZSLPC (Cheraghian et al., 2020) | 3.86 | – | 10.37 | 4.14 | – | 8.49 |
| 3DGenZ (Michele et al., 2021) | 6.46 | – | 18.33 | 6.03 | – | 15.86 |
| ZSLPC (Cheraghian et al., 2019) | 9.97 | – | 18.70 | 9.52 | – | 17.16 |
| ShapeLLM$*$ (Qi et al., 2024) | 0.88 | 0.28 | 0.99 | 1.49 | 1.35 | 1.70 |
| OpenAD-PointNet++ (Nguyen et al., 2023) | 13.53 | 3.97 | 16.40 | 11.29 | 2.41 | 13.88 |
| OpenAD-DGCNN (Nguyen et al., 2023) | 11.15 | 3.84 | 13.86 | 8.04 | 1.58 | 9.85 |
| IAGNet (Yang et al., 2023) | 16.16 | 19.07 | 23.92 | 14.36 | 16.90 | 21.73 |
| LASO (Li et al., 2024) | 22.41 | 15.90 | 30.22 | 20.06 | 8.80 | 26.84 |
| Ours-Qwen | 24.43 | 23.90 | 35.45 | 26.25 | **29.5** | **41.57** |
| Ours-Phi | **30.43** | **29.36** | **47.78** | **27.25** | 27.87 | 39.04 |

points), and mAcc$^c$ (mean accuracy over all classes). However, unlike OpenAD, which includes the "none" category in the calculation of metrics, we only compute the 36 affordance types, excluding "none," as it has little comparative significance. For a comprehensive evaluation versus existing methods, we additionally assess each instance across the entire dataset. The specific evaluation metrics over all **instances**: mIoU$^i$ (mean IoU over all instance data), mAcc$^i$ (mean accuracy of points over all instance data), mPrec$^i$ (mean precision of points over all instance data), mRec$^i$ (mean recall of points over all instance data), mAP$^i_{50}$ (mean average precision at 50% intersection over union).

## 4.2 EXPERIMENT RESULTS

### 4.2.1 COMPARISON RESULTS

**3D-ADLLM vs. Other Models.** Table 1 demonstrates that our 3D-ADLLM achieves superior performance across both full and partial view tasks, as well as on all three evaluation metrics. Notably, 3D AffordanceLLM significantly outperforms the runner-up model (LASO) in terms of mIoU, with improvements of 8.02% and 7.19% on the full and partial view tasks, respectively. Compared to OpenAD, which predicts regions based on a fixed set of affordance labels, our method utilizes long-context understanding and reasoning for segmentation. In experiment results, our method surpasses OpenAD in terms of mIoU 16.9% (full-view) and 15.96% (partial-view) separately across 18 affordance types. Additionally, for metrics over all instance, we surpass the sota model (LASO) 23.38% (full-view) and 24.93% (partial-view) in mAP$_{50}$. The comparison results on close-set detection can be found in Appendix Sect. A.2.

### 4.2.2 OUT-OF-DISTRIBUTION RESULTS

The test in out-of-distribution (ood) datasets is essential to assess the generalization capability of the model. Thus, we constructed a new test dataset consisting of approximately 559 entries by filtering out some combinations of affordance-object that already existed in our IRAS dataset from the AffordPose dataset (Jian et al., 2023). Compared to existing datasets, this new dataset includes different types of affordances as well as unique affordance-object pairs, such as *(twist, faucet), (lever, faucet), (press, dispenser)*, etc. As is shown in Table 3, our approach achieved the best zero-shot performance on this ood data.

## 4.3 ABLATION STUDY

**Effects of Different Components.** To investigate the effectiveness of each component in 3D-ADLLM, we conduct experiments with different variants of 3D-ADLLM. In particular, we compare 2 different implementations: (1) w/o PC removes the pre-trained weights $f_{PB}$ and $f_{AFD}$, directly training our 3D-ADLLM; (2) w/o UL removes the sample unbalanced factor. As is shown in Table 6,

Table 2: Zero-shot Open-vocabulary detection results on over **all instances.**

| | Method | mIoU[i] | mAcc[i] | mPrec[i] | mRec[i] | mAP[i]$_{50}$ |
|---|---|---|---|---|---|---|
| Full-view | OpenAD-PointNet++ | 3.46 | **74.59** | 11.84 | 5.84 | 0.02 |
| | OpenAD-DGCNN | 3.79 | 74.42 | 11.13 | 6.67 | 0.04 |
| | LASO | 20.47 | 71.47 | 37.95 | 34.93 | 2.42 |
| | 3D-ADLLM (ours) | **30.28** | 70.66 | **40.89** | **55.93** | **27.80** |
| Partial-view | OpenAD-PointNet++ | 2.17 | 71.97 | 5.64 | 3.74 | 0.02 |
| | OpenAD-DGCNN | 2.08 | 72.00 | 6.65 | 3.40 | 0.02 |
| | LASO | 11.46 | **72.14** | 32.70 | 16.49 | 0.70 |
| | 3D-ADLLM (ours) | **28.72** | 68.28 | **41.71** | **47.73** | **25.63** |

Table 3: Zero-shot Open-vocabulary detection results on AffordPose data over **all instances.**

| Method | mIoU[i] | mAcc[i] | mPrec[i] | mRec[i] | mAP[i]$_{50}$ |
|---|---|---|---|---|---|
| OpenAD-PointNet++ | 7.61 | 65.13 | 22.47 | 13.01 | 0.37 |
| OpenAD-DGCNN | 8.02 | 66.76 | 15.83 | 13.52 | 0.39 |
| LASO | 34.49 | **77.12** | **56.04** | 37.88 | 8.40 |
| 3D-ADLLM (ours) | **36.33** | 74.79 | 55.46 | **46.80** | **36.33** |

the performance of 3D-ADLLM drops significantly without either of these components. Notably, the most substantial performance degradation with about 6% occurs in mIoU when the PC module is removed. UL is also critical for our framework. Once it is removed, the performance, there is a noticeable reduction in the model's performance.

**Effects of Different Backbones.** As shown in Table 1, we experimented with different LLM backbones to evaluate the effectiveness of our framework. Specifically, we chose Phi-3.5-mini-instruct (Abdin et al., 2024) and Qwen2-1.5B (Yang et al., 2024) as the LLM backbone. In terms of experimental results, Phi outperforms Qwen in the full-view setting. However, in the partial-view setting, the performance of Phi shows no significant difference compared to Qwen. Based on these findings, 3D-ADLLM adopts Phi as the default LLM backbone. In addition to testing different LLM backbones, we also explored different point encoders. Table 4 summarizes the performance of ULIP2 (Xue et al., 2024) and Uni3D (Zhou et al., 2023) as point encoders, while ULIP2 obtained slightly better mean accuracy.

Table 4: The efforts with different point encoder $f_{\mathrm{pe}}$ in 3D-ADLLM.(Full-View)

| $f_{\mathrm{pe}}$ | mIoU[c] | Acc[c] | mAcc[c] |
|---|---|---|---|
| ULIP2 | **30.43** | **29.36** | 47.78 |
| Uni3D | 30.26 | 26.21 | **48.16** |

**Effects of Different Learning Objectives.** We define the Affordance Region Ratio (Arr) as $p_{\mathrm{aff}}/p_{\mathrm{cloud}}$, representing the proportion of affordance regions relative to the point cloud. In the IRAS task, the average Arr is approximately 18%. However, for specific categories like "pull" and "listen," it is around 5%, while for "wear" it reaches about 40%. Variations in Arr across different predictions lead to class imbalances. Dice Loss, a segmentation loss function, measures the similarity between predictions and ground truth. Unlike Binary Cross-Entropy Loss (BCE), which focuses on pixel-level differences, Dice Loss emphasizes global region similarity, making it more effective for handling data imbalance. As shown in Table 5, the model utilizing Dice Loss achieves superior mIoU metrics in both seen and unseen settings. Table 5 demonstrates that while the exclusive application of Dice Loss yields a marginal improvement on unseen data, it does not perform as well on seen data when compared to the combined usage of Dice Loss and BCE Loss.

Table 5: The comparison results regarding different settings of loss.(full-view)

|  | Openset-mIoU$^c$ | Closeset-mIoU$^c$ |
|---|---|---|
| DICE & BCE | 30.43 | **42.35** |
| DICE | **31.00** | 38.65 |
| BCE | 15.99 | 31.14 |

Table 6: Results of 3D-ADLLM variants with removing different components.(full-view)

| Model | mIoU$^c$ | Acc$^c$ | mAcc$^c$ |
|---|---|---|---|
| 3D-ADLLM | **30.43** | **29.36** | **47.78** |
| 3D-ADLLM $_{w/o\ PC}$ | 24.82 | 20.54 | 36.73 |
| 3D-ADLLM $_{w/o\ UL}$ | 25.35 | 26.84 | 40.69 |

## 4.4 QUALITATIVE RESULTS

As shown in Fig. 4, our model demonstrates the capacity to accurately comprehend object affordance given the complex reasoning instruction. It is noteworthy that even when dealing with small affordance components, such as the switch of faucet, our model still exhibits decent ability. Moreover, our 3D-ADLLM surpasses other models by employing a multi-stage training strategy that facilitates knowledge transfer and extraction of world knowledge from LLMs. For example, when identifying areas on a chair that can take a seat Fig. 4 (e) or areas that can wrap around a cup Fig. 4 (g), our model significantly outperforms other models.

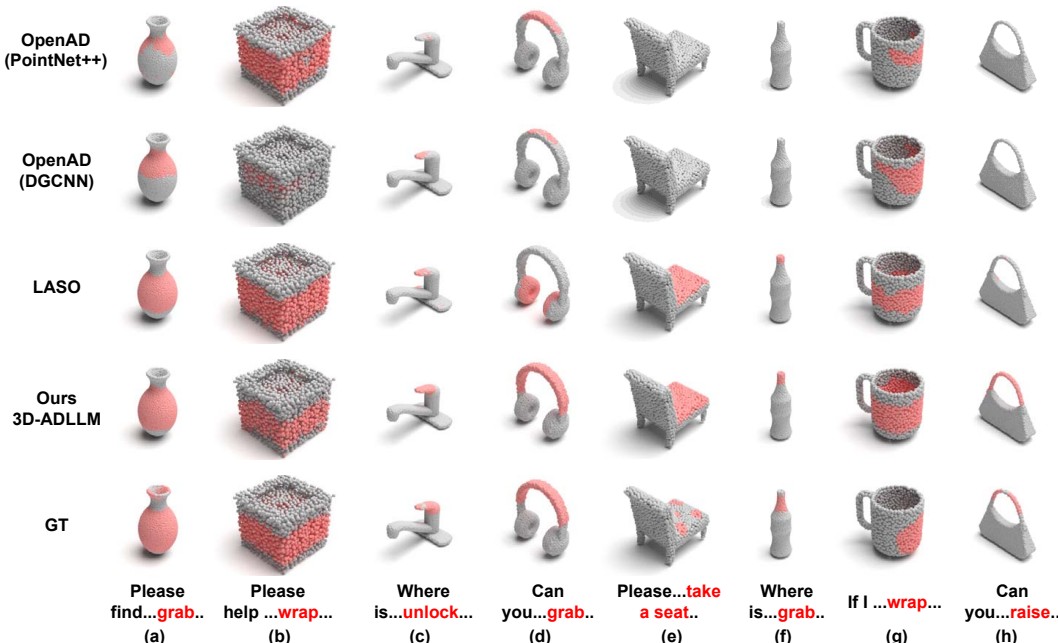

Figure 4: The visualization results of our 3D-ADLLM compared with others.

## 5 CONCLUSION

In this work, we reformulate the traditional affordance detection paradigm into *Instruction Reasoning Affordance Segmentation* (IRAS) task, enabling open-world affordance detection. Then, we propose the multi-stage learning strategy with a novel defined Referring Object Part Segmentation (ROPS) task to extract general segmentation knowledge to affordance detection. Finally, we accordingly proposed the 3D-AffordanceLLM (3D-ADLLM), firstly injecting LLM into 3D affordance perception, a framework designed for query reasoning affordance segmentation in 3D open scenarios. Experimental results demonstrate the effectiveness of 3D-ADLLM, we hope our work can shed new light on the direction of affordance detection in open-world scene in the future.

## 6 ACKNOWLEDGEMENT

We would like to thank the reviewers for their constructive comments. This study is supported by Shenzhen College Stability Support Plan (Grant No. GXWD20220817144428005), National Natural Science Foundation of China (Grant No. 62406092), National Natural Science Foundation of China (Grant No. U24B20175), Shenzhen Science and Technology Program (Grant No. KJZD20240903100017022), Research on Efficient Exploration and Self-Evolution of APP Agents & Embodied Intelligent Cerebellum Control Model and Collaborative Feedback Training Project (Grant No. TC20240403047).

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

# A APPENDIX

## A.1 BASELINE MODELS DETAILS

We compare our method with the following recent methods for zero-shot learning in 3D point clouds: ZSLPC (Cheraghian et al., 2019), TZSLPC (Cheraghian et al., 2020), and 3DGenZ (Michele et al., 2021). For these baselines, we change their original text encoders with CLIP and retain the same settings in OpenAD (Nguyen et al., 2023). Furthermore, we incorporate two affordance detection works (IAGNet (Yang et al., 2023) and LASO (Li et al., 2024)) to provide a more comprehensive comparison of our approach. For IAGNet (Yang et al., 2023), an affordance detection method that utilizes paired image-point cloud data as input. To tailor IAGNet (Yang et al., 2023) to our requirements, we seamlessly integrate a language model in place of its original image backbone, while maintaining the rest of its architecture unchanged. ShapeLLM-7B (Qi et al., 2024) is a large-scale point cloud model that accepts point cloud and natural language inputs and possesses grounding capabilities. Consequently, we leverage its grounding abilities to perform zero-shot detection and calculate masks for comparison.

## A.2 COMPARISON RESULTS ON CLOSE SET

In this work, we primarily focus on enhancing affordance detection capabilities in open-world scene. However, our model still performs well in closed-set affordance detection tasks. As is shown in Fig. 7, Fig. 8, our 3D-ADLLM achieves optimal performance on nearly all metrics in both the over-all-classes and over-all-instances settings.

Table 7: Main results of 3D-ADLLM compared with other methods on close-set detection over all class.

| Method | Full-view | | | Partial-view | | |
| --- | --- | --- | --- | --- | --- | --- |
| | $mIoU^c$ | $Acc^c$ | $mAcc^c$ | $mIoU^c$ | $Acc^c$ | $mAcc^c$ |
| Point Transformer (Zhao et al., 2021) | 41.26 | – | 67.03 | 40.51 | – | 65.34 |
| PointNet++ (Qi et al., 2017) | 41.26 | – | **68.14** | 41.10 | – | **66.74** |
| DGCNN (Wang et al., 2019) | 42.09 | – | 61.47 | 41.93 | – | 63.12 |
| OpenAD-PointNet++ (Nguyen et al., 2023) | 40.17 | 38.61 | 66.83 | 40.44 | 38.92 | 65.84 |
| OpenAD-DGCNN (Nguyen et al., 2023) | 41.17 | 35.71 | 59.17 | 39.87 | 35.15 | 59.27 |
| IAGNet (Yang et al., 2023) | 40.04 | 35.12 | 53.05 | 41.24 | 34.68 | 52.58 |
| LASO (Li et al., 2024) | 41.31 | 35.02 | 53.96 | 40.11 | 35.21 | 52.68 |
| 3D-ADLLM | **42.85** | **41.84** | 66.35 | **41.92** | **43.40** | 61.93 |

Table 8: The performance of close set affordance detection over **all instances**.

| | Method | mIoU[i] | mAcc[i] | mPrec[i] | mRec[i] | mAP$_{50}^{i}$ |
|---|---|---|---|---|---|---|
| Full-view | OpenAD-PointNet++ | 28.34 | 64.11 | 33.91 | 61.45 | 5.12 |
| | OpenAD-DGCNN | 26.98 | 65.94 | 34.38 | 54.76 | 4.77 |
| | LASO | 44.43 | **83.80** | **62.73** | 60.25 | 21.13 |
| | 3D-ADLLM (ours) | **46.29** | 81.24 | 57.90 | **64.27** | **46.38** |
| Partial-view | OpenAD-PointNet++ | 29.50 | 63.26 | 35.21 | 61.34 | 6.77 |
| | OpenAD-DGCNN | 17.07 | 67.11 | 27.96 | 30.15 | 1.87 |
| | LASO | 43.35 | **82.31** | 60.27 | 59.57 | 20.85 |
| | 3D-ADLLM (ours) | **44.06** | 79.64 | 56.23 | **64.21** | **46.60** |

## A.3 DATA ANALYSIS

### A.3.1 THE DETAILED SETTINGS OF FULL VIEW AND PARTIAL VIEW.

Building the IRAS dataset based on the 3D AffordanceNet (OpenAD) Dataset (Nguyen et al., 2023). We retained the settings for both full-view point cloud and partial-view point cloud.

Full-view: Given an object as 3D point cloud without knowing the affordances supported by the object, the full-shape affordance estimation task aims to estimate the supported affordance type and predict the point-wise probabilistic score of affordance.

Partial-view: in real-world application scenarios, we can only expect partial view of 3D shapes, represented as partial point cloud. Therefore, another important task we are concerned with is to estimate the affordance from partial point cloud.

### A.3.2 THE VISUALIZATION OF DATA ANALYSIS.

## A.4 TRAINING DETAILS

We constructed our IROS dataset based on the PartNet dataset and aim to acquire general segmentation knowledge through the pre-training phase. To balance training costs with model performance, we selectively sampled a subset of the data based on categories to obtain general segmentation knowledge for objects.

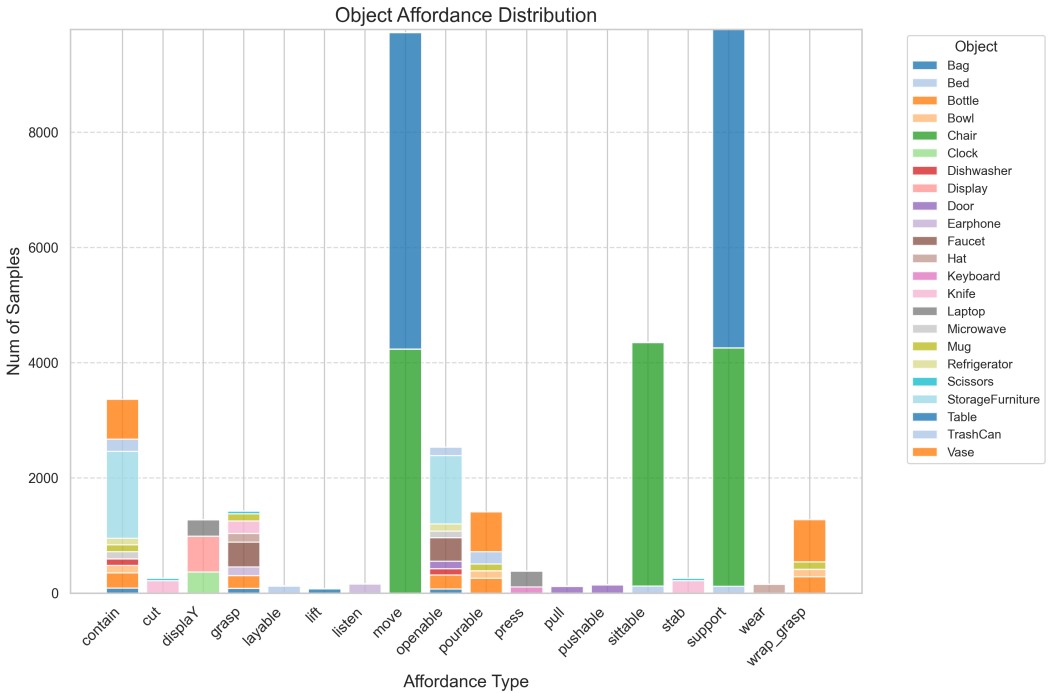

Figure 5: The analysis of IRAS task.

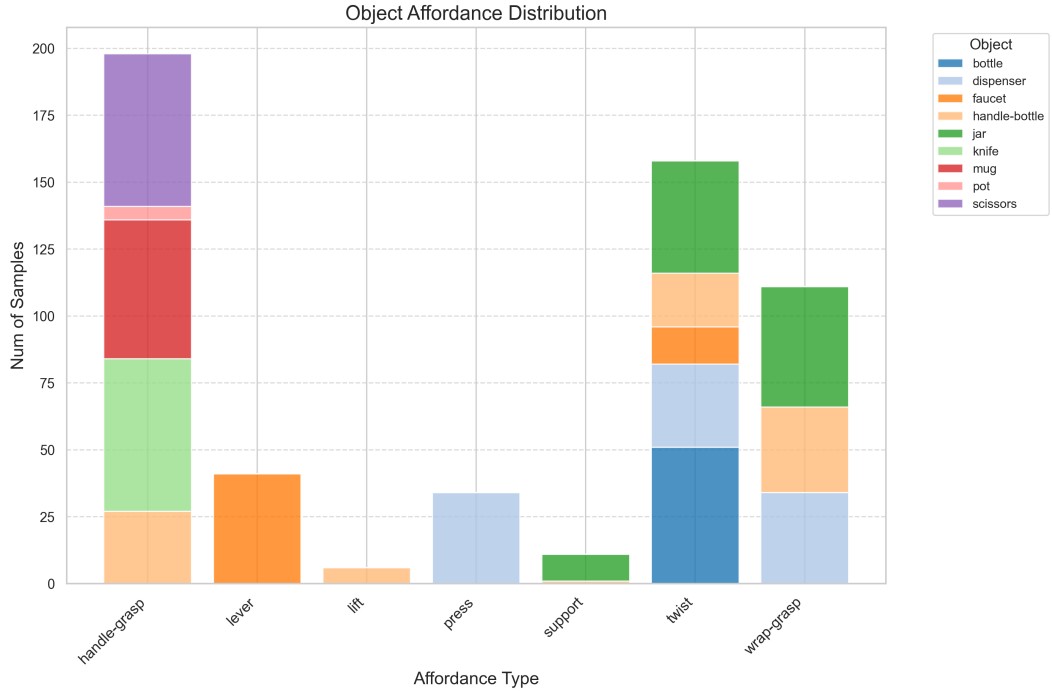

Figure 6: The analysis of extensive test dataset in sec. 4.2.2.

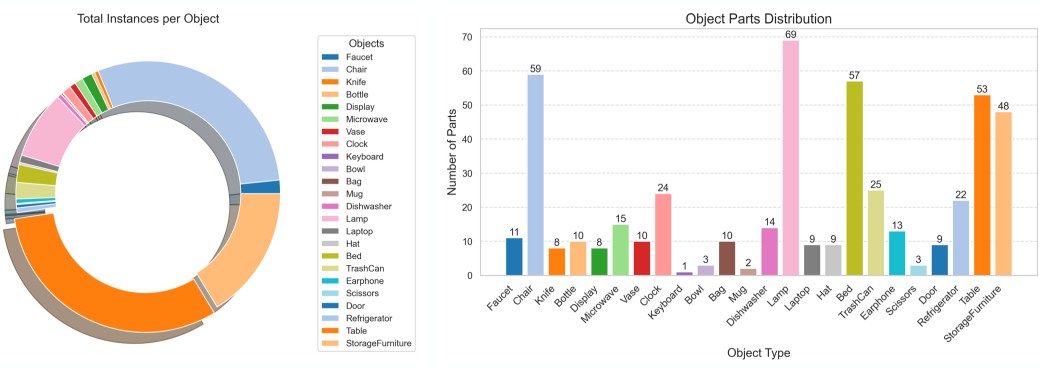

Figure 7: The analysis of ROPS task.

