# SUPPLEMENTARY MATERIALS

## 1 RESULTS ON EACH AFFORDANCE CATEGORY

According to the Table 1, We found that our model achieved the highest mIoU in 12 categories. However, in 3 categories, other models outperformed ours by approximately 7% in mIoU. The largest difference was observed in the category slice, where other models exceeded our performance by about 50%. Upon analysis, we believe that IAGNet and OpenAD-PN2 overfit to certain categories in zero-shot open-vocabulary affordance detection, leading to significant advantages in a few specific categories. However, I believe our training strategy is still superior. Compared to OpenAD, we achieve over a 10% lead in 13 categories, with 6 categories exceeding 30%. The largest margin is approximately 40%.

Table 1: The open vocabulary mIoU evaluation results for each affordance category on full-view setting. The best results of our methods are in bold, while best mIoU of other methods are marked with ∗. We believe the results ∗ perform well due to the model overfitting to certain categories.

| Category | Ours | IAGNet | LASO | OpenAD-PN2 |
|---|---|---|---|---|
| push | **0.3326** | 0.0440 | 0.2853 | 0.2440 |
| drag | 0.0387 | 0.0010 | 0.0451∗ | 0 |
| unlock | **0.2543** | 0.0672 | 0.1565 | 0.0320 |
| demonstrate | **0.4381** | 0.1981 | 0.3619 | 0.0670 |
| accommodate | **0.1473** | 0.0289 | 0.0945 | 0.0010 |
| grab | 0.3031 | 0.3376∗ | 0.0267 | 0.0350 |
| hear | 0.4282 | 0.1786 | 0.4482 | 0.4890∗ |
| wrap | **0.4759** | 0.3798 | 0.4553 | 0.2090 |
| pour | **0.4326** | 0.3597 | 0.4147 | 0.2160 |
| slice | 0.0296 | 0.0587 | 0.0379 | 0.5430∗ |
| jab | **0.4205** | 0.0012 | 0.2887 | 0 |
| raise | 0.0656 | 0.1473 | 0.0291 | 0 |
| take a seat | **0.6015** | 0.2558 | 0.2434 | 0.2960 |
| bear | 0.3437 | 0.0696 | 0.3590∗ | 0 |
| reposition | 0.2272 | 0.4146∗ | 0.0665 | 0 |
| thumb | **0.4121** | 0.0096 | 0.3942 | 0 |
| rest | **0.2850** | 0.2274 | 0.2164 | 0.1510 |
| clothe | **0.2421** | 0.1076 | 0.1095 | 0.1450 |

## 2 MORE QUALITATIVE RESULTS

**Qualitative results on real-world partial-view point cloud**

We collected some other partial-view point cloud in real world for affordance detection. The visualization results are shown in 1.

**Multi-affordance results**

As illustrated in Figure 2, an object can possess multiple reasonable affordances. For instance, the mug depicted in Figure 2 demonstrates four types of affordances: pourable, wrap, grab, and accommodate (from left to right).

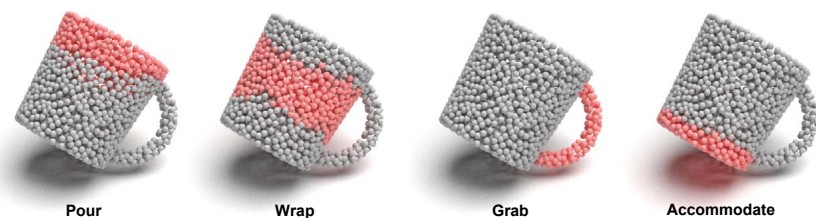

Figure 1: Qualitative results on real-world partial-view point cloud.

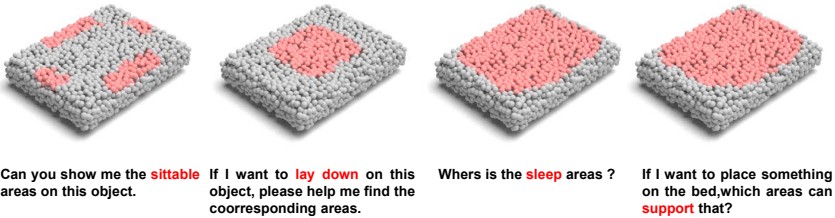

Figure 2: Different affordance types of an object

Additionally, a single part or region on an object may have multiple reasonable affordances. For instance, we explored the different affordances of a bed, as shown in the Figure 3. We found that there are some regions that support both sleeping, and also have attributes such as support, laying down, and sittable.

Figure 3: Different affordance types with same regions on an object.

**Fine-grained Results**

A part of an object often carries specific affordance meanings; for example, the handle of a knife can be used for grasping. Moreover, the region of the knife handle intended for grasping should not be limited to a small area; therefore, our model tends to predict an entire part. Of course, our model can also predict fine-grained regions,in Figure 4: the tip of a knife—jab, the tip of scissors—stab, the rim of the mug-pour,door-open.

**OOD visualization results.**

As shown in Figure 5, our model demonstrates promising performance on the OOD data.

## 3  ADDITIONAL EXPERIMENTS

**Effects of different pre-training datasets** We conduct the experiment that the first stage is trained on 3D AffordanceNet dataset. Compared to pre-training 3D AffordanceNet, our model benefits from IROS pretraining, acquiring general segmentation knowledge and effectively transferring it to affordance detection. As the table shows, our method achieves a 6.13% increase in mIoU, 9.24% in Acc, and 13.76% in mAcc.

**Detailed comparison with OpenAD**

As described in Line 400–407 of our paper regarding the evaluation metrics, we have modified the calculation methods for mIoU, Acc, and mAcc. Unlike OpenAD, which includes the background

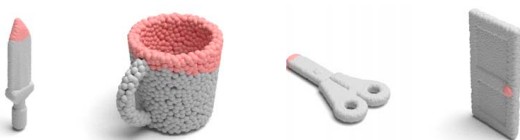

Figure 4: Fine-grained Results

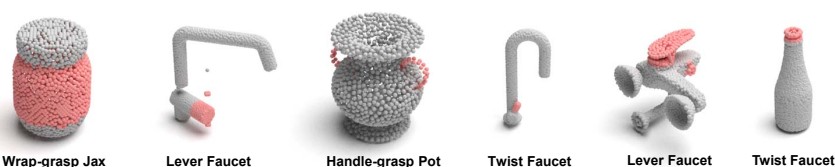

**Wrap-grasp Jax**  **Lever Faucet**  **Handle-grasp Pot**  **Twist Faucet**  **Lever Faucet**  **Twist Faucet**

Figure 5: Visualization of OOD.

Table 2: The efforts with different pre-training strategy in 3D-ADLLM.

| Method | mIoU | Acc | mAcc |
|---|---|---|---|
| Pretrain-3DAffordanceNet | 24.30 | 20.12 | 34.02 |
| 3D-ADLLM | **30.43** | **29.36** | **47.78** |

Table 3: Main results of 3D-ADLLM on zero-short open vocabulary detection. The result is calculated over all classes(∗: included "none").

| Method | Full-view | | | Partial-view | | |
|---|---|---|---|---|---|---|
| | mIoU | Acc | mAcc | mIoU | Acc | mAcc |
| TZSLPC∗ | 3.86 | 42.97 | 10.37 | 4.14 | 42.76 | 8.49 |
| 3DGenZ∗ | 6.46 | 45.47 | 18.33 | 6.03 | 45.24 | 15.86 |
| ZSLPC∗ | 9.97 | 40.13 | 18.70 | 9.52 | 40.91 | 17.16 |
| OpenAD-PointNet++ | 13.53 | 3.97 | 16.40 | 11.29 | 2.41 | 13.88 |
| OpenAD-PointNet++∗ | 15.19 | 46.68 | 20.69 | 13.01 | 44.90 | 18.37 |
| OpenAD-DGCNN | 11.15 | 3.84 | 13.86 | 8.04 | 1.58 | 9.85 |
| OpenAD-DGCNN∗ | 12.97 | 46.45 | 18.23 | 9.88 | 44.09 | 14.53 |
| Ours-Phi | 30.43 | 29.36 | 47.78 | 27.25 | 27.87 | 39.04 |
| Ours-Phi∗ | **32.18** | **72.05** | **52.13** | **29.05** | **70.36** | **43.61** |

as the "none" category in metric calculations, we chose to exclude the "none" category, as it holds no practical significance. Our evaluation data exhibits a clear class imbalance, with the "none" category accounting for a large proportion. However, the "none" category lacks real significance and its disproportionate presence affects the evaluation of our results. Therefore, we chose to exclude the "none" category when calculating the metrics. In addition, we also update the evaluation results with the "none" category added in Figure 3. Experimental results demonstrate that, in the original metric calculations of OpenAD, our model still maintains optimal performance.

**Ours vs. LLM as Part-classfier**

In our method, the LLM extracts its intrinsic world knowledge and outputs a `<AFF>` multimodal representation to predict the affordance mask. In addition, we conducted an ablation experiment by freezing the LLM, where it was used solely as a part classifier. As shown in Table 3, our method effectively extracts prior knowledge of the LLM in affordance, resulting in a significant performance improvement compared to the approach in which the LLM is used solely as a part classifier.

Table 4: Comparison with LLM as a part classifier vs ours 3D-ADLLM. Zero-short open-vocabulary results on IRAS dataset all classes.

| Method | mIoU | Acc | mAcc |
|---|---|---|---|
| LLM-Classifier | 24.42 | 20.83 | 42.43 |
| Ours-Phi | **30.43** | **29.36** | **47.78** |

**Efforts of Different Components with Detailed Experiments**

Basic Baseline A: Use the LLM to encode only text features, inputting them into the decoder along with point features encoded by a standard backbone, like PointNet++.

Pretraining Variant B: Use the proposed pretraining task, ROPS, train variant A and then fine-tune A on the IRAS task.

Backbone Variant C: Replace the point encoder with the same point segmentation backbone (Point Transformer).

Dual Encoder Variant D: Introduce a second point encoder, initially using PointNet++, which outputs point features to the LLM along with text tokens. The LLM's output is then fed into the decoder from Variant C.

Current Approach E: Test the full model with the proposed AFF token integration and two encoders to measure the combined effectiveness of all components.

Comparison with previous approaches F: The fairest comparison to the previous approaches is to remove the pretrain stage and replace your segmentation backbone with PointNet++.

Table 5: Detailed Ablation Study on Full-view dataset with metrics over all classes.

| Method | mIoU | Acc | mAcc |
|---|---|---|---|
| A(PN2) | 20.88 | 21.36 | 37.41 |
| B(PN2) | 25.39 | 21.81 | 43.82 |
| C-A(PT) | 19.75 | 20.95 | 36.86 |
| C-B(PT) | 24.41 | 20.83 | 42.43 |
| E (PN2) | 27.92 | 29.20 | 47.40 |
| E (PT) | **30.43** | **29.36** | **47.78** |
| D | 24.16 | 25.04 | 40.21 |
| F | 23.40 | 24.17 | 35.43 |

(1)Setting A: By comparing Method A with OpenAD, we find that replacing the language model with an LLM enhances the performance of OpenAD to a certain extent (mIoU:13.53 $\rightarrow$ 20.88), which also reflects the impact of introducing a large language model.

(2) Setting A vs.Setting B: By replacing the point backbone network with the standard PointNet++ backbone, the mIoU improves from 20.88 to 25.39, Acc increases from 21.36 to 21.81, and mAcc rises from 37.41 to 43.82. This demonstrates that our multi-stage training strategy effectively enhances model's performance on the task.

(3) Setting C: Comparing C-A(PT) and C-B(PT) with A(PN2) and B(PN2), C-A(PT) and C-B(PT) have better performance than A(PN2) and B(PN2) respectively (C-B vs. B(A): mIoU: 24.41 → 25.39, Acc: 20.83 → 21.81, mAcc: 42.43 → 43.82). This shows that that the performance improvement stems not from the backbone choice but from the proposed multi-training strategy.

(4) Setting E: In our 3D-ADLLM, the point encoder $f_p e$ is frozen which outputs point features to the LLM along with text tokens. Therefore, I believe that the comparison between E and B better highlights the role of the <AFF> token marker. Comparing E(PN2) and E(PT) with B(PN2) and B(PT), the <AFF> token extracts affordance-related prior information from the LLM, resulting in an increase of mIoU by 2.53% (PN2), 6.02% (PT).

(5) Setting D: Initializeing the trainable PointNet++ to replace the frozen $f_{pe}$ PointBert-ULIP2. By comparing setting D with setting E (PT), we find that in our method, initlizing $f_{pe}$ with a model trained with point cloud and text alignment, and freezing $f_{pe}$ for further training is effective and improves performance. Experimental results also validate our conclusion.

(6) Setting F: By removing the pretrain stage and replacing the point backbone with PointNet++, we give the fairest comparison. Compared to LASO: mIoU: 22.41 → 23.40, Acc: 15.90 → 24.17, mAcc: 30.22 → 65.43, our method outperforms previous approaches.