# OpenReview forum: "3D-AffordanceLLM: Harnessing Large Language Models for Open-Vocabulary Affordance Detection in 3D Worlds"
_ICLR.cc/2025/Conference — ICLR 2025 Poster_

### Official Review · Reviewer_UMiE · 2024-11-02

**Soundness:** 3
**Presentation:** 2
**Contribution:** 3
**Rating:** 6
**Confidence:** 3

**Summary:**

The authors propose a method to enable open-vocabulary affordance detection in the 3D world via utilising the knowledge from Large Language Models. A new framework named 3D-AffordanceLLM is proposed. The authors propose a multi-stage training strategy to overcome the problem of the scarcity of 3D affordance datasets. A new pre-training task is therefore designed, named Referring Object Part Segmentation and is to train the model to have the capability of general recognition and segmentation. Subsequently, the model is fine-tuned with Instruction Reasoning Affordance Segmentation. Experiments show the effectiveness of the proposed method.

**Strengths:**

1. I like the idea of using the knowledge from Large Language Models for open-vocabulary detection. It is sensible as LLM may contain knowledge about the world and can have good generalisation in terms of open-vocabulary.

2. The design of the architecture is novel. An <AFF> token is added into the language model, which connects to the affordance prediction. It is a natural design.

**Weaknesses:**

1. For organisation of the paper, it would be better if the authors can focus more on the current work rather than using several paragraphs to describe the development of previous works some of which are not closely related.

2. For experiments, it would be better if the authors can explain more about the numbers. For example, in the last row of Table 2, it is a bit strange why mAcc is so low compared with the rest, while the other numbers are the highest. Are these metrics consistent?

3. Both the pre-training and evaluation of the model are conducted on synthetic datasets, but it is an important problem how the model can work in the real world as there can be a large domain gap between the real world scenarios and the synthetic datasets. I am not fully convinced unless the authors can show their training on synthetic datasets can have a good zero-shot capability in real world scenarios. It would be better if the authors can provide some results in real world scenarios.

**Questions:**

I would suggest the authors to address the questions raised in the weakness section in the discussion period.


*** Post - discussion

The authors' comments have addressed some of my concerns, and I therefore decide to change raise my score to be 6.

---

### Official Review · Reviewer_ovue · 2024-11-03

**Soundness:** 2
**Presentation:** 2
**Contribution:** 2
**Rating:** 6
**Confidence:** 3

**Summary:**

The paper’s key advantage lies in its integration of large language models (LLMs) into affordance segmentation, leveraging their semantic richness to enhance understanding and address limitations of fixed affordance labels in prior methods, thus increasing adaptability in open-world settings. The proposed task, Instruction Reasoning Affordance Segmentation (IRAS), combines LLM-encoded text features with point features, with a pretraining step on part segmentation to address data scarcity.

However, IRAS closely resembles prior affordance tasks, particularly Language-guided Affordance Segmentation on 3D Objects (LASO), which raises questions about its novelty. Additionally, the methodology shows minimal distinction from existing work, with pretraining as the main difference, which does not provide strong novelty. Another limitation is the lack of rigorous comparative experiments to demonstrate clear advantages over previous approaches, leaving key aspects, such as the contributions of LLM features and pretraining, unclarified.

The paper also suffers from poor organization and inconsistent notation, which hinder readability. While additional experiments could strengthen validation, the limited novelty and unclear contributions reduce the impact of the work. Therefore, I recommend rejection.

**Strengths:**

1. **Motivation**: the strength of this paper is its motivation to leverage large language models (LLMs) for enhancing semantic comprehension in 3D affordance detection. Unlike previous methods that are limited by a fixed set of affordance labels, this approach uses an LLM to interpret complex, instruction-based queries, enabling the model to recognize affordances in a more flexible and context-aware manner. By incorporating LLMs, the model can process nuanced instructions, overcoming the constraints of rigid label sets and allowing it to generalize affordances in open-world settings.

2.  **Pretraining**: The authors address the challenge of limited data by introducing a pretraining stage on part segmentation, which enhances the model's ability to capture part-level features even with a smaller affordance dataset. This pretraining strategy not only compensates for data scarcity but also provides the model with a foundation to learn more transferable features, contributing to improved performance in affordance segmentation.

**Weaknesses:**

1. **Limited Novelty Regarding the Proposed Task**: The authors introduce a task, Instruction Reasoning Affordance Segmentation (IRAS), aiming to address perceived limitations in existing 3D affordance detection approaches. However, IRAS shares substantial overlap with previous work, particularly with the Language-guided Affordance Segmentation on 3D Objects (LASO) task[1]. Both IRAS and LASO involve generating a mask over functional parts of a 3D object point cloud based on a question or instruction. Specifically, each query in both tasks consists of an object type and an affordance type, with the objective of inferring the functional part relevant to the query. To clarify the distinctiveness of IRAS, the authors should elaborate on how their queries are generated, detailing the heuristics and rationale behind their construction, and the specific differences between IRAS and LASO, and any innovations that set IRAS apart as a unique contribution to the field. Without this, the similarities between IRAS and LASO bring into question the novelty of IRAS as a distinct task.
2. **Limited Innovation in Methodology**: To address these challenges, the authors integrate a large language model (LLM) into a point cloud segmentation framework, using the LLM to process instructions and produce text features along with point cloud features for affordance mask prediction. However, the design involving two separate point encoders is confusing. The authors may claim that incorporating multimodal text and point cloud inputs allows the LLM to capture semantic information better, potentially enhancing segmentation performance. However, given the limited dataset size, it is unclear if sufficient point-text alignment can realistically be achieved. Furthermore, this design lacks supporting experiments to validate its effectiveness. Aside from a part segmentation pretraining step for extracting part-based features, the approach doesn't show a significant difference from LASO, which uses text features from LLM and point cloud features from a point backbone to decode affordance masks. This makes the contribution of pretraining alone insufficient as a distinguishing factor for novelty. The author needs to explain the core difference between your method and LASO.
3. **Lack of Rigorous Experimental Validation**: though the proposed method shows performance gains over previous approaches, the effectiveness of the proposed design is not convincingly demonstrated due to concerns about the experimental setup and baseline data, as noted in the questions. A key issue is the lack of ablations to demonstrate the extent to which the improvements stem from the pretraining stage, the large-scale pretrained backbone, or the method itself. The following baseline experiments could strengthen the validation. The authors are encouraged to prioritize experiments based on their assessment of each module's importance, conducting analyses to substantiate the effectiveness of different components and explaining which aspects they consider most critical, and why.
    - **Basic Baseline A**: Use the LLM to encode only text features, inputting them into the decoder along with point features encoded by a standard backbone, like PointNet++. This approach would closely follow LASO’s baseline method.
    - **Pretraining Variant B**: Use the proposed pretraining task, ROPS, train variant **A** and then fine-tune **A** on the IRAS task. This would help isolate the contribution of pretraining compared to **A** and show its impact on performance. In your current experiments, you simply remove the pretraining step (as shown in **Table** 6). This makes it difficult to determine whether the improvements over LASO are due to the proposed `AFF` token or the use of the Point Transformer segmentation backbone. The Without additional experiments, it’s challenging to draw convincing conclusions about the source of the performance gains.
    - **Backbone Variant C**: Replace the point encoder with the same point segmentation backbone (Point Transformer) used in this work, to determine the extent to which the performance improvement is due to the backbone choice rather than the proposed method.
    - **Dual Encoder Variant D**: Introduce a second point encoder, initially using PointNet++, which outputs point features to the LLM along with text tokens. The LLM’s output is then fed into the decoder from Variant **C**. This setup would reveal how much the proposed `AFF` token contributes to performance.
    - **Current Approach E**: Test the full model with the proposed `AFF` token integration and two encoders to measure the combined effectiveness of all components.
    - **Comparison with previous approaches F**: The fairest comparison to the previous approaches is to remove the pretrain stage and replace your segmentation backbone with PointNet++.
4. **Inadequate Clarity and Organization in the Paper**: The methodology section lacks clear logical flow, and inconsistent notation makes it challenging to follow. The experimental setup is insufficiently detailed, and numerous errors in descriptions further reduce clarity. Improvements in notation consistency, methodology organization, and experimental description are needed to enhance the overall clarity and coherence of the proposed approach.

[1]Yicong Li, Na Zhao, Junbin Xiao, Chun Feng, Xiang Wang, and Tat-seng Chua. Laso: Language-
guided affordance segmentation on 3d object. CVPR 2024

**Questions:**

- **Q1**: In **Figure 1**, the example question is ambiguous. Given a prompt like “something I can use to drink water” for a mug, most people would interpret this as referring to the whole mug, not specifically the handle or the body. In **Figure 2**, the questions are much clearer and directly target specific parts. Can you explain how your model infers that the handle of a mug is the intended affordance in such an ambiguous scenario in **Figure 1**? Is any specific design in your method to handle ambiguity or resolve cases with multiple possible interpretations? You could provide a range of example queries, from ambiguous to specific, to illustrate how the model addresses each type of query and manages interpretational differences.
- **Q2**: In **Line 210**, the explanation and organization of the point encoder section are unclear and inconsistent. Why is the subscript for $f_{pe}$ lowercase, while $f_{PB}$ is uppercase? According to the projector section, the feature X is fed into the LLM. However, in **Figure 2**, it appears that feature X originates from the "point backbone" rather than the "point encoder," as mentioned in **Line 210**. Could you clarify this inconsistency?
- **Q3**: In **Lines 289-297**, the explanation is unclear, and the notations are inconsistent. The symbols $\textbf{f}$, $f_{pb}$, and $\textbf{p}_{clouds}$ do not match the earlier sections or the figures. Could you provide a clearer and more cohesive explanation?
- **Q4**: In **Figures 2 and 3**, the "b" stage is inconsistent. In **Figure 2**, the point cloud feature input into the LLM is encoded by a "trainable point backbone," while in **Figure 3**, it is processed by a "frozen point encoder." Could you clarify this pipeline, ensuring consistent notations and descriptions?
- **Q5**: In **Line 341**, how is the ground truth text obtained?
- **Q6**: In **Line 353**, the formula for $L_{mask}$ appears incorrect. The summation should be outside the formula. It seems that your query already includes the affordance type, meaning the affordance type is known beforehand. Therefore, this task can be treated as a binary classification problem, as opposed to a multi-class setting. In OpenAD, they maximize the cosine similarity between points and the target affordance type, making it a multi-class classification problem. Given this setup, I’m curious about the role of the affordance class weights in your binary classification loss. Could you clarify how these affordance class weights are used in this binary classification context?
- **Q7**: In **Table 1**, you claim the experiment is zero-shot and open-vocabulary and you seem to reference results from TZSLPC, 3DGenZ, and ZSLPC in OpenAD[2]’s **Table 1**. If your setup is identical to OpenAD’s **Table 1**, which evaluates 37 classes, why do you mention evaluating only 18 valid classes in **Line 402**? Could you clarify your experimental setup?
- **Q8**: In **Table 1**, how is the accuracy (Acc) computed? In OpenAD’s **Table 1**, all reported accuracies are over 40%, yet here, your best result is only 29%, and the two OpenAD variants show values around 3.9%. Given that your reported mIoU and mAcc for the two OpenAD variants are closer to those in OpenAD, one would expect similar Acc values. Where do these reported accuracy results for the two OpenAD variants come from, and why aren’t you referencing the accuracies for TZSLPC, 3DGenZ, and ZSLPC, even though you reference other metrics for these models?
- **Q9**: In **Table 1**, could you specify what “full view” and “partial view” refer to in your experimental setup?
- **Q10**: In **Table 2**, could you further clarify the difference between the mAcc in **Table 2** and the mAcc in **Table 1**? It’s somewhat understandable that "over all classes" and "over all instances" might differ, but why does, for example, the mAcc for OpenAD-PointNet++ differ so significantly from 16.4 in **Table 1** to 74.59 in **Table 2**?
- **Q11**: In **Tables 2 and 3**, why is the performance on out-of-distribution (OOD) unseen object data higher than on the test set of your primary dataset in **Table 2**? Adding visual examples could help clarify this discrepancy.
- **Q12**: In **Tables 4 and 6**, the experiment setting is highly unclear, on what dataset were the experiments conducted, and did you use the full view or partial view setting?

[2]Toan Nguyen, Minh Nhat Vu, An Vuong, Dzung Nguyen, Thieu Vo, Ngan Le, and Anh Nguyen.Open-vocabulary affordance detection in 3d point clouds. IROS 2023.

---

### Official Review · Reviewer_PzNX · 2024-11-03

**Soundness:** 3
**Presentation:** 2
**Contribution:** 3
**Rating:** 6
**Confidence:** 4

**Summary:**

This paper proposes to solve open-world affordance prediction by integrating LLM into the pipeline. The pipeline start with pre-training the point cloud backbone and affordance mask decoder on part segmentation task conditioned on text. Then, the LLM is fine-tuned with affordance discovery on specific tasks. During inference, the LLM will take the complex text description and point cloud tokens encoded from a fixed encoder and output the affordance token <AFF>, conditioned on which the pre-trained affordance mask decoder will predict the affordance mask on the point cloud. Compared with zero-shot learning methods on point cloud, this framework outperforms in terms of accuracy and generalization ability.

**Strengths:**

1. Leveraging common sense from LLM for 3D affordance grounding is well-motivated, which could potentially help downstream open-vocabulary manipulation.
2. The main challenging of using LLM for 3D affordance prediction is handling the multimodal input and output format as segmentation mask. This paper propose a neat way to tokenize point features and convert the text prediction to a condition for affordance discovery.

**Weaknesses:**

1. While this framework connects the LLM with point encoding and mask decoding and could handle complex text instructions, my major concern is about the way of combining LLM with affordance prediction. In the IRAS fine-tuning stage, the output of the LLM is directly supervised by text targets $y_{txt}$, combined with the fact that the point backbone and mask decoder are pre-trained on text-conditioned part segmentation task, it seems that the current framework is using LLM as just a part classifier and use decoder as a text-conditioned part segmentation network. The visualization also shows that this system tend to predict the whole part as the affordance mask. This could limit the method's application on more fine-grained affordance grounding task, where only a small region of part should be activate. Besides, in this way, the benefit of using LLM is just handling complex instruction and convert it into segmentation text instead of providing detailed prior on affordance (e.g., which region provides a firm grasping). I believe there could be more integrated way of combining LLM to fully leverage the priors from it.
2. The design of the architecture also shows that the current function of LLM is just handling complex text and output text condition for segmentation. For example, in Figure 3, the text condition of the Mask Decoder is the name of part during pre-training, whereas in IRAS fine-tuning it is replace with the output of LLM.
3. The figure 2 and Figure 3 have misalignment on the use of two point cloud networks. I seems that there is a typo in Figure 2 since the Point Backbone is "specifically tailored for segmentation tasks".
4. Minor: The quality of Figure 2 is not good. The fonts of some text seems wrong.

**Questions:**

1. Could you please show more qualitative results on the affordance prediction to see whether it has tendency for predicting the whole part?
2. The use of two point cloud networks need to be clarified since there are misalignment between two figures and descriptions in the paper.
3. Could you explain more on the two "Projector" in Figure 3? Are they the same? It seems that they have different input and output domain.
4. What does the text targets $y_{txt}$ in IRAS fine-tuning refer to?

---

### Official Review · Reviewer_oFkQ · 2024-11-03

**Soundness:** 2
**Presentation:** 2
**Contribution:** 2
**Rating:** 5
**Confidence:** 4

**Summary:**

The paper focuses on point cloud affordance segmentation. It adapts the LISA model to the 3D affordance domain. Specifically, given a text prompt requesting an affordance and a 3D point cloud, it fine-tunes a Large Language Model (LLM) to predict a specialized affordance token, which is decoded with point cloud embeddings to predict the segmentation mask on the point cloud. It is first trained on the 3D part dataset PartNet (pointwise part label) and further on the annotated 3D AffordanceNet dataset (pointwise affordance label with questions). It appears to be evaluated on the OpenAD dataset and claims to outperform previous works.

**Strengths:**

1. It extends LISA to the 3D domain. It's interesting to find LLMs can also predict the next token for 3D tasks.

2. It makes point queries more open-vocabulary. The paper supports asking for affordances in natural-language questions, making the input question more flexible.

3. Dataset annotation effort. The authors annotate the 3D AffordanceNet dataset (point labels) with questions.

**Weaknesses:**

1. Not open-world. The method cannot generalize to unseen 3D object parts, as the current 3D part dataset PartNet covers a limited category of objects with few instances. The proposed method is 3D part data dependent, which is a significant bottleneck. It is too preliminary to claim it works in a 3D open-world scene. In contrast, existing works on open-world part segmentation of 3D point clouds, such as PartSLIP, address this issue.

2. No detailed table on per-class evaluation results. The dataset is unbalanced, and per-class results should be reported.

3. No qualitative results on real-world partial-view point cloud segmentation.

4. No multi-affordance experiments. A single part or region on an object may have multiple reasonable affordances, which is crucial in an open-vocabulary scenario. Related experiments are missing here.

5. Ablation: What if the first stage is also trained on 3D AffordanceNet?

6. No robotics experiments. Affordance is always about interaction, which differentiates it from general part semantics. There are no real-world experiments to show if these better predictions lead to higher robotic manipulation success rates. Many papers introduce real-world robotic experiments to demonstrate learned affordance: SayCan, Where2Explore, OpenAD, SAGE, etc.

**Questions:**

The paper doesn’t clearly state which dataset is used for evaluation. Inferred from L400, it may be 3D AffordanceNet (OpenAD annotated). Please make it clearer in the paper.

Moreover, there is a clear discrepancy between the quantitative results in OpenAD and this paper. Please explain.

Editing Suggestions:

L109: transferr → transfer

L313: Defination → Definition

L366: typo: projector

L370: Refering → Referring

L534: enableing → enabling

L708: no space in baselines

I suggest the authors use an LLM to perform a grammar check.

---

### Official Review · Reviewer_uifM · 2024-11-04

**Soundness:** 3
**Presentation:** 3
**Contribution:** 2
**Rating:** 6
**Confidence:** 4

**Summary:**

The paper introduces 3D-AffordanceLLM, a novel framework for open-vocabulary affordance detection in 3D point clouds that reformulates traditional affordance detection as an Instruction Reasoning Affordance Segmentation (IRAS) task. Moving beyond fixed label-based approaches, the framework combines Large Language Models (LLMs) with a custom affordance decoder to generate segmentation masks from natural language queries. To address the challenge of limited 3D affordance datasets, the authors propose a multi-stage training strategy that first extracts knowledge from general segmentation tasks through a novel Referring Object Part Segmentation (ROPS) pre-training task, followed by fine-tuning with IRAS. This approach leverages the rich world knowledge and reasoning capabilities of LLMs while equipping the model with general recognition and segmentation capabilities at the object-part level. The method demonstrates significant improvement, achieving approximately 8% increase in mIoU on open-vocabulary affordance detection tasks, showcasing its effectiveness in bridging the gap between natural language understanding and 3D affordance detection.
.

**Strengths:**

- The problem of affordance detection in 3D is important for robotics applications and human-robot interaction1
- The technical approach is sound and demonstrates meaningful improvements over baselines
- The paper is well-written and clearly structured

**Weaknesses:**

My primary concerns are:

- The motivation for using LLMs is not sufficiently justified. The paper argues that traditional methods are limited by fixed label sets and short-text detection, but recent works like CLIP-based approaches have shown success in open-vocabulary detection [1].
- The paper lacks proper discussion of its relationship to similar concurrent work, particularly AffordanceLLM [2]. The architectural similarities and differences should be explicitly addressed. The primary difference is the feature encoder (PointBERF for point cloud vs OWLViT for 2D images).

[1] Gen Li, Deqing Sun, Laura Sevilla-Lara, Varun Jampani. One-Shot Open Affordance Learning with Foundation Models. CVPR 2024.
[2] Shengyi Qian, Weifeng Chen, Min Bai, Xiong Zhou, Zhuowen Tu, Li Erran Li. AffordanceLLM: Grounding Affordance from Vision Language Models. CVPRW 2024.

Additional comments which do not need to be addressed

L366 projcetor -> projector

**Questions:**

See my primary concerns at weakness.

---

### Meta-Review · Area_Chair_qos4 · 2024-12-20

**Metareview:**

This paper proposes a method called 3D-AffordanceLLM for open-vocabulary affordance detection in 3D point clouds, which is framed as an Instruction Reasoning Affordance Segmentation (IRAS) task, analogous to the 2D counterpart LISA. The proposed 3D-AffordanceLLM integrates large language models (LLMs) into 3D affordance perception by designing a specific token to encode knowledge from LLMs, which is then input to the decoder to generate affordance masks. To address the data shortage challenge in 3D affordance datasets, the authors first pre-train their model on the 3D part segmentation dataset, PartNet, using the objective of Referring Object Part Segmentation (ROPS). The model is then fine-tuned on the 3D AffordanceNet dataset with the objective of IRAS.

Initial reviewer concerns focused on several aspects of the paper, including:

- The motivation for using LLMs (uifM)
- Similarity to concurrent work, such as AffordanceLLM [CVPRW 2024] (uifM)
- Generalizability to unseen parts (oFkQ)
- Lack of experimental results, including per-class performance, multi-affordance detection, and visualizations (oFkQ, PzNX)
- Lack of Implementation details, such as evaluation datasets and settings (oFkQ, UMiE)
- Design limitations in integrating LLMs with affordance prediction for fine-grained affordance segmentation (PzNX)
- Limited novelty in the task and method (ovue)
- Insufficient experimental validation, and request for additional experiments, such as an ablation study on the first stage and real-world robotic experiments (oFkQ)

The authors provided detailed responses to the reviewers' concerns, including clarifications and additional experiments. After the rebuttal and the subsequent AC-reviewer discussion, three reviewers (ovue, UMiE, PzNX) voted to borderline accept the paper, while two reviewers (oFkQ, uifM) voted to borderline reject it. Notably, reviewers oFkQ and uifM did not actively follow up during the author response period.

Overall, the paper’s contributions lie in its novel part segmentation pre-training strategy and the integration of LLMs with the affordance decoder. However, there are some weaknesses, particularly the similarity of the task to previous work, such as LASO, and the incremental contributions to object affordance detection. Furthermore, the paper lacks sufficient validation on open-world affordance grounding.

Despite these weaknesses, the AC believes the contributions outweigh the limitations, particularly the potential utility of the pre-training strategy for the 3D affordance segmentation field. Given the subjective nature of the paper’s contributions, the AC believes it is appropriate to let the community assess its potential impact. The authors have also done a good job addressing the reviewers' concerns.

Therefore, the AC recommends accepting this paper, with the suggestion that the authors include details of the real-world robotic experiments and provide additional examples on open-world affordance grounding in the final version.

**Additional Comments On Reviewer Discussion:**

Initial reviewer concerns focused on several aspects of the paper, including:

- The motivation for using LLMs (uifM)
- Similarity to concurrent work, such as AffordanceLLM [CVPRW 2024] (uifM)
- Generalizability to unseen parts (oFkQ)
- Lack of experimental results, including per-class performance, multi-affordance detection, and visualizations (oFkQ, PzNX)
- Lack of Implementation details, such as evaluation datasets and settings (oFkQ, UMiE)
- Design limitations in integrating LLMs with affordance prediction for fine-grained affordance segmentation (PzNX)
- Limited novelty in the task and method (ovue)
- Insufficient experimental validation, and request for additional experiments, such as an ablation study on the first stage and real-world robotic experiments (oFkQ)

The authors provided detailed responses to the reviewers' concerns, including clarifications and additional experiments. After the rebuttal and the subsequent AC-reviewer discussion, three reviewers (ovue, UMiE, PzNX) voted to borderline accept the paper, while two reviewers (oFkQ, uifM) voted to borderline reject it. Notably, reviewers oFkQ and uifM did not actively follow up during the author response period.

The AC believes the contributions outweigh the limitations, particularly the potential utility of the pre-training strategy for the 3D affordance segmentation field. Given the subjective nature of the paper’s contributions, the AC believes it is appropriate to let the community assess its potential impact. Therefore, the AC is inclined to accept the paper. However, the AC would not object if the paper were to be rejected.

---

### Decision · Program_Chairs · 2025-01-22

Accept (Poster)